# Impact of COVID-19 on the Stage and Treatment of Endometrial Cancer: A Cancer Registry Analysis from an Italian Comprehensive Cancer Center

**DOI:** 10.3390/cancers17162686

**Published:** 2025-08-18

**Authors:** Francesca Roncaglia, Lucia Mangone, Francesco Marinelli, Isabella Bisceglia, Maria Barbara Braghiroli, Valentina Mastrofilippo, Fortunato Morabito, Antonia Magnani, Antonino Neri, Lorenzo Aguzzoli, Vincenzo Dario Mandato

**Affiliations:** 1Epidemiology Unit, Azienda USL-IRCCS di Reggio Emilia, 42122 Reggio Emilia, Italy; francesca.roncaglia@ausl.re.it (F.R.); francesco.marinelli@ausl.re.it (F.M.); isabella.bisceglia@ausl.re.it (I.B.); mariabarbara.braghiroli@ausl.re.it (M.B.B.); 2Obstetrics and Gynecological Oncology, Azienda USL-IRCCS di Reggio Emilia, 42122 Reggio Emilia, Italy; valentina.mastrofilippo@ausl.re.it (V.M.); lorenzo.aguzzoli@ausl.re.it (L.A.); vincenzodario.mandato@ausl.re.it (V.D.M.); 3Gruppo Amici Dell’Ematologia Foundation-GrADE, 42123 Reggio Emilia, Italy; fortunato.morabito@grade.it; 4Operations Management Unit, Azienda USL-IRCCS di Reggio Emilia, 42122 Reggio Emilia, Italy; antonia.magnani@ausl.re.it; 5Scientific Directorate, Azienda USL-IRCCS di Reggio Emilia, 42122 Reggio Emilia, Italy; antonino.neri@ausl.re.it

**Keywords:** endometrial neoplasm, COVID-19, stage, grading, time to surgery, treatment, Italy

## Abstract

The COVID-19 pandemic has impacted the diagnosis and treatment of endometrial cancers. This population-based study evaluates the pandemic’s effects on endometrial cancer stage, delays between diagnosis and surgery, and kind of surgical approach. We analyzed all endometrial cancers diagnosed between 2017 and 2023, recorded in the Reggio Emilia Cancer Registry, northern Italy, across three periods, namely pre-COVID (2017–2019), COVID (2020–2022), and post-COVID (2023). During these three periods, stages I and II showed no significant variation (*p* = 0.66; *p* = 0.58), while a significant increase was observed in stage III (*p* < 0.05), and a slight increase was noted in grade 3 tumors. Furthermore, a significant decrease in laparotomies (*p* < 0.01) and an increase in laparoscopies (*p* < 0.05) were noted. The time to surgery (the number of days between diagnosis and surgery) increased, shifting from 1–2 months to over 2–3 months. Delays were more pronounced among older women and those diagnosed with stage IV disease.

## 1. Introduction

Endometrial cancer (EC) is one of the most common gynecologic malignancies, with 420,368 incident cases and 97,723 deaths reported worldwide each year [1]. The lifetime risk of developing EC for women is approximately 3%. Over recent decades, the incidence of EC has increased substantially—potentially rising by 132% globally—primarily due to an aging population and the growing prevalence of diabetes and obesity, both of which are recognized as significant risk factors for EC in both developed and emerging countries [2].

Endometrial cancer primarily affects postmenopausal women, with a median age at diagnosis of 61 years. However, there has been a notable increase among younger women over the past three decades, with incidence in women under 50 years rising from 4 to 7 cases per 100,000 [3]. Although no standardized screening tests exist for EC, it is often diagnosed at an early stage due to symptomatic presentation [4,5]. The most common initial symptom is abnormal vaginal bleeding, which typically prompts gynecologic evaluation, leading to diagnostic hysteroscopy (DH) and targeted endometrial biopsy. DH is considered the gold standard for assessing endometrial pathology and has progressively replaced traditional dilation and curettage [6]. Early-stage diagnosis is associated with favorable outcomes, with a 5-year survival rate of approximately 80% [7]. However, advanced-stage EC presents significant therapeutic challenges and is associated with higher mortality [8].

In early 2020, the onset of the COVID-19 pandemic led to an unprecedented healthcare crisis, with hospitals being rapidly restructured to accommodate critically ill patients with severe respiratory infections [9]. Routine medical activities, including cancer screenings and diagnostic tests, were suspended or postponed to prioritize the management of COVID-19 patients [10]. This unfavorable situation also extended to surgical interventions [11], with non-urgent surgeries deferred [12]. In gynecologic oncology, timely intervention is crucial, particularly for EC—where delays of over six weeks from diagnosis to surgery may adversely impact survival outcomes [12,13]. In Italy, the first wave of COVID-19 was in the period between March and June 2020, with a second wave in December: the lockdown periods continued throughout 2021. Therefore, 2022–2023 were considered as the complete resumption of healthcare activities, both diagnostic and therapeutic.

This study aims to assess the impact of the COVID-19 pandemic on the management of EC and to determine whether these disruptions compromised oncological outcomes.

## 2. Materials and Methods

### 2.1. Sources of Information

This study included all cases of EC diagnosed between 2017 and 2023 and treated in the Obstetrics and Gynecology Unit at Azienda USL–IRCCS, a Comprehensive Cancer Center in Reggio Emilia, Italy. These cases were recorded by the Reggio Emilia Cancer Registry (RE-CR) and classified as topography C54 according to the International Classification of Diseases for Oncology, Third Edition (ICD-O-3) [14]. Data from the RE-CR, which is primarily sourced from pathology reports, hospital discharge records, and mortality statistics, were integrated through record linkage with the gynecology department dataset. Variables, such as stage, grade, diagnostic procedures, type of treatment, and surgery, were added. The RE-CR covers a population of approximately 532,000 inhabitants and is recognized as a high-quality cancer registry, reporting a high percentage of microscopic confirmation (98.8%) for EC and a low proportion of cases identified by death certificate only (<0.1%) [15].

The RE-CR collects data to generate statistics on incidence, mortality, prevalence, and survival for the resident population and demographic subgroups, as mandated by Italian Law No. 29 of 22 March 2019, which governs cancer registries in Italy. Under this law, cancer registries are exempt from requiring informed consent. This study’s epidemiological analyses, based on the RE-CR data, were approved by the Provincial Ethics Committee of Reggio Emilia (Protocol no. 2014/0019740 of 4 August 2014).

### 2.2. Analysis of Data

Endometrial cancer cases were grouped into three time periods for comparison, namely pre-COVID-19 (2017–2019), COVID-19 (2020–2022), and post-COVID-19 (2023).

The study includes all endometrial infiltrating tumors registered at the Reggio Emilia Cancer Registry between 2017 and 2023. All cases were included in the analyses, since it is a population-based study.

In the Reggio Emilia Cancer Registry area, there are few private clinics, while the public system is very active, providing free care to all residents, including foreigners. The few private clinics present are, however, affiliated with the National Health System: therefore, the Cancer Registry receives the same information (morphology reports and discharge forms) as patients admitted to public facilities (which in our case account for approximately 99% of the total).

The distribution of EC cases by age, morphology, stage, grade, treatment, and diagnostic imaging was analyzed by period of diagnosis. To evaluate the time to surgery (TTS), the interval (in days) between diagnosis and treatment (surgery) was calculated, with the mean and standard deviation reported for each period. The date of diagnosis (which coincides with the date of incidence) means the date of the first definitive histological report was used; in its absence, the date of hospital admission was used.

Delays in surgery were categorized as ≤30, 31–60, 61–90 days, and >90 days, excluding 131 cases where the date of surgery coincided with the date of diagnosis (meaning the date of diagnosis as mentioned above). A multiple regression analysis was performed to evaluate factors influencing TTS (dependent variable), using age, year of treatment, and stage. Regression coefficient (β), 95% confidence intervals (CI), and *p*-values were reported for statistically significant results. All analyses were performed using STATA 16.1 SE software (StataCorp, College Station, TX, USA). Finally, to better observe the changes in incidence in the periods examined, a trend was added with the incidence rates standardized by age, stratified by year.

## 3. Results

In the period 2017—2023, 543 cases of endometrial cancer were diagnosed, with an average patient age at diagnosis of 66.6 years (Table 1). Of these, 228 EC cases were diagnosed in the pre-COVID period (2017—2019), 242 were diagnosed during the COVID period (2020—2022), and 73 cases were diagnosed in the COVID-free period (2023). The majority of tumors (58%) were identified in women aged ≥65 years. Furthermore, 77.9% of cases were diagnosed as stage I, 3.7% as stage II, 9.8% as stage III, and 3.7% as stage IV; 32.6% were grade 1, 36.8% were grade 2, and 15.1% were grade 3. Imaging data showed that most patients (69.8%) underwent a computed tomography (CT) scan, either alone or (5.3%) in combination with magnetic resonance imaging (MRI). Surgical approaches included laparoscopic surgery (LPS) in 70.9% of cases and laparotomy (LPT) in 19.3%.

The distribution of stage and grade by period is reported in Table 2. Stages I and II showed no significant variation (*p* = 0.66; *p* = 0.58), while a significant increase was recorded in stage III (7.5%, 9.5%, and 17.8%, respectively, in the pre-COVID, COVID, and COVID-free periods; *p* < 0.05). The increase in grade 3 tumors (15.4%, 13.6%, and 19.2% in the three periods considered, respectively) was not statistically significant (*p* = 0.70). The distribution by type of treatment in the three periods (Table 3) shows a significant decrease in laparotomies already evident during COVID (30.3%, 11.6%, and 11.0%, respectively; *p* < 0.01) and a significant increase in laparoscopies (60.1%, 78.1%, and 80.8%, respectively; *p* < 0.05).

The time to surgery (TTS, days between diagnosis of the tumor and surgery) is reported in Table 4. Across the three periods considered, a statistically significant decrease was observed in the proportion of patients who underwent surgery within both 30 days of diagnosis (10.1%, 3.7%, and 1.4%, respectively; *p* < 0.01) and 60 days (38.6%, 19.4%, and 6.9%, respectively; *p* < 0.01). Conversely, a statistically significant increase in TTS was observed for surgeries performed after 60 days (22.8%, 29.8%, and 37.0%, respectively; *p* < 0.05) and after 90 days (7.5%, 23.1%, and 20.5%, respectively; *p* < 0.01).

Multivariate analysis (Table 5) identified significant associations with time to surgery: overall, longer TTS was observed with increasing patient age (β = 0.92 [95% CI: 0.38–1.44]), diagnosis period (β = 22.79 [95% CI: 14.67–30.91]), and cancer stage (β = 9.87 [95% CI: 3.55–16.19]). Using the categorized variables and compared to the COVID period, a statistically significant reduction in TTS was observed in the pre-COVID period (β = −19.63 [95% CI: −31.31 to −7.95]) and a similarly significant increase was observed in the post-COVID period (β = 31.60 [95% CI: 13.68–49.53]). Regarding stage, a significantly increased TTS was observed only for stage IV (β = 48.80 [95% CI: 23.15–74.45]).

The trend in the incidence of endometrial cancer over the period considered shows a stability in rates during the COVID period with an increase in 2022 and a return to previous values in 2023 (Figure 1).

## 4. Discussion

The study, conducted using population-based data from the Reggio Emilia Cancer Registry and clinical data from a Comprehensive Cancer Center, highlights critical trends and challenges that arose during the COVID-19 period. Between 2017 and 2023, 543 patients with endometrial cancer were included in the study. The analysis of the three periods—2017–2019 (pre-COVID-19), 2020–2022 (COVID-19), and 2023 (COVID-19-free)—showed that during the COVID period, there was an increase in stage III tumors, suggesting that delays in diagnostic procedures led to a gradual increase in tumor size and lymph node involvement. Although not statistically significant, a modest increase in grade 3 tumors was also observed across the three periods.

The closure of operating rooms and the shift of many hospital activities toward the COVID emergency also had a high impact on surgical procedures, with a decrease in laparotomies and an increase in laparoscopic surgeries. For the same reasons, TTS was also affected: in general, waiting times between diagnosis and surgery were within 30 days (10% of cases) or 60 days (39% of cases). During the COVID pandemic, delays increased, with a rise in the proportion of patients who waited more than 60 days (30% in the COVID period and 37% in the COVID-free period) and more than 90 days (23% and 21%, respectively).

Adjusting for age, the COVID-free period showed a significant increase in TTS (*p* < 0.01) compared to the pre-COVID period, which persisted in subsequent years. As for stage, patients in stages II and III did not show changes in TTS compared to those in stage I; only patients with stage IV tumors experienced a delay in surgical treatment.

### 4.1. Results in the Context of Published Literature

During the COVID period, a decline in diagnoses was recorded, largely linked to the decrease in non-urgent diagnostic procedures [16]; endometrial biopsies decreased by 40% [17], although no significant reductions were observed in EC surgeries [18]. In our study, diagnostic hysteroscopies (DH) decreased by 16% during the COVID period but returned to initial levels by 2023.

Various factors, such as the reorganization of hospital services, reductions in non-urgent treatments, and patient fears of contracting COVID-19, likely contributed to these decreases in DH, though without a significant impact on new cancer diagnoses, which in our series remained stable—as already reported in Dutch, Canadian [18,19], and Italian [20,21] studies. An American study, however, reported a 35% reduction in EC diagnoses during 2020 compared to 2019 [22], and another reported a 19.1% decline from March 2020 to December 2020 compared with equivalent pre-COVID months [23]. Other works have addressed the problem of patients’ fear of going to hospital during the pandemic [24], also due to the risk of opportunistic infections [25,26].

The pandemic’s impact on endometrial cancer stage remains a subject of ongoing research, with studies reporting varying trends in distribution. Several studies have documented a decline in early-stage diagnoses and a concurrent increase in advanced stages during the pandemic [20,27]. Garrett et al. reported a decrease in stage I diagnoses from 86.1% pre-pandemic to 71.7% during the lockdown [27], while Bogani et al. observed a reduction from 74.3% to 71.7% during the same period [20].

Our study reports a significant increase in stage III only, from 7.5% in the pre-COVID period to 17.8% in the post-COVID period. Similar findings have been reported in an Irish study, where stage III cases increased from 12% to 19% [28], and a multicenter Italian study, which noted an increase from 12.8% to 14.3% [20]. Concerning stage IV, some studies have reported modest increases during the pandemic: Bogani et al. observed an increase in stage IV cases from 4.7% to 6.8% during the pandemic [20], and Garrett et al. reported a rise from 2.1% to 5.5% [27]; in contrast, neither our studies nor other reports [18,21] found a significant increase in stage IV endometrial cancers during the COVID period.

During the COVID-19 pandemic, therapeutic approaches underwent notable changes. The decrease in laparotomy and the rise in laparoscopy likely reflect technological advancements, increasing surgeon expertise, and the influence of scientific society guidelines [29], even amidst the challenges posed by the COVID-19 pandemic. Similarly, surgical interventions for various gynecologic tumors decreased during this period in different institutions [7,30]. In some centers, early-stage endometrial cancer was managed with hormonal therapy as a temporary measure, and surgeries were postponed [31]. However, in our center and other Italian centers, treatment protocols remained consistent, with no significant deviation from established practices [32]. Despite concerns over the increased SARS-CoV-2 infection risks associated with aerosol-generating procedures, which led to a reduction in laparoscopic surgery (LPS) in some centers [33,34], our study and others observed an increase in the use of LPS [31]. This divergence may reflect regional variations in healthcare policies and resource allocation during the pandemic.

One of the most significant impacts of the pandemic was the increase in the time interval from diagnosis to surgical treatment (TTS). A reduction from 70.9 days to 49.3 days was observed in U.S. public hospitals, while private hospitals experienced fewer delays [31,35]. In Canada, the delay between biopsy and surgery remained relatively stable, increasing slightly from 56 days during COVID-19 to 58 days in the pre-COVID-19 period [16].

In our study, TTS was exceptionally prolonged during 2021–2022: multivariable analysis revealed that TTS increased with age, particularly for patients diagnosed at stage IV. Some studies suggest that prolonged surgical wait times can adversely affect EC prognosis [13]; a delay exceeding six weeks has been associated with worse overall survival in early-stage type I EC [36], while recent evidence indicates that disease-free survival is negatively impacted in advanced-stage disease when surgery is delayed beyond six weeks [37].

### 4.2. Strengths and Weaknesses

This study presents several strengths related to the fact that it is based on population-level data; therefore, there is no selection bias, and it includes 2023 incidence data. The quality of the data is high, as the number of missing cases is extremely low—partly because all cases are reviewed by a tumor board, which reduces treatment heterogeneity [31]. An important limitation is that this is the work of a single cancer registry; however, during the COVID emergency, the level of hospitalizations and deaths was high in northern Italy, and, in general, all levels of care, participation in cancer screening, diagnostics, and treatment are very similar across northern Italy. The regions of northern Italy share the same incidence of cancer and the same survival rates. Therefore, the study results can easily be extended to all areas of northern Italy characterized by the same levels of care.

Another limitation of the study is the lack of information on comorbidity, which could impact the management of women, especially older women. Unfortunately, this information is not easily accessible in Italy, and, therefore, consulting these databases would require specific authorization.

## 5. Conclusions

Our study showed that, during the COVID-19 period, there was an increase in more advanced endometrial tumors and a prolongation of waiting times between diagnosis and surgery. The study results were shared with gynecologists: perhaps the fact that gynecologists changed their approach during the pandemic (performing a greater number of laparoscopies than laparotomies), even in older women and those with more advanced tumors, might justify publishing this study. Although it refers to a single center, the findings are certainly generalizable, at least to northern Italy, which has a healthcare system with very similar diagnostic and therapeutic approaches.

## Figures and Tables

**Figure 1 cancers-17-02686-f001:**
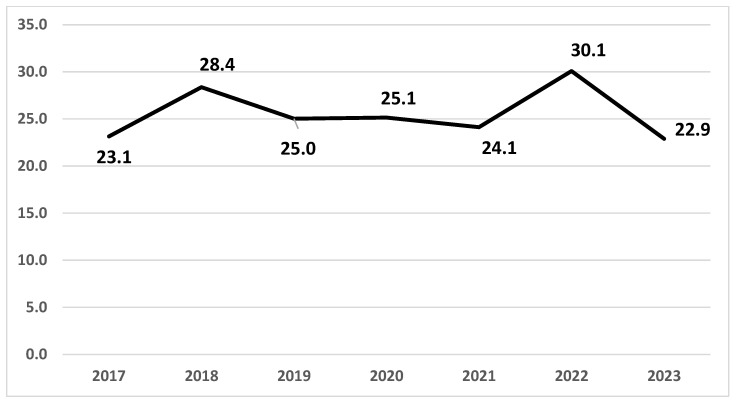
Reggio Emilia Cancer Registry. Endometrial cancer 2017–2023. Standardized incidence trend.

**Table 1 cancers-17-02686-t001:** Reggio Emilia Cancer Registry. Endometrial cancers, years 2017–2023. Distribution of cases by years of diagnosis, age, stage, grade, diagnosis, and treatment.

	*n*	%
Overall	543	
Years of diagnosis		
2017–2019	228	
2020–2022	242	
2023	73	
Age at diagnosis (brackets)		
<40	4	0.7
40–64	224	41.3
≥65	315	58.0
Stage		
I	423	77.9
II	20	3.7
III	53	9.8
IV	20	3.7
Unknown	27	5.0
Grade		
G1	177	32.6
G2	200	36.8
G3	82	15.1
Unknown	84	15.5
Diagnostic imaging		
MRI #	51	9.4
CT #	379	69.8
MRI # + CT #	29	5.3
XR #	29	5.3
PET #	1	0.2
Other/unknown	54	9.9
Surgical approach		
Laparotomy	105	19.3
Laparoscopy	385	70.9
Vaginal hysterectomy	16	3.0
Laparotomy + laparoscopy	23	4.2
Unknown	14	2.6

# MRI: magnetic resonance imaging; CT: computed tomography; XR: X-ray; PET: positron emission tomography.

**Table 2 cancers-17-02686-t002:** Reggio Emilia Cancer Registry. Distribution of endometrial cancers by stage and grade across the three diagnosis periods.

	2017–2019	2020–2022	2023		Total
Number of cases	228	242	73		543
Stage	*n* (%)	*n* (%)	*n* (%)	*p*-value	
I	179 (78.5)	192 (79.3)	52 (71.2)	0.66	423
II	8 (3.5)	8 (3.3)	4 (5.5)	0.58	20
III	17 (7.5)	23 (9.5)	13 (17.8)	<0.05	53
IV	11 (4.8)	8 (3.3)	1 (1.4)	0.17	20
Unknown	13 (5.7)	11 (4.6)	3 (4.1)	—	27
Grade	*n* (%)	*n* (%)	*n* (%)	*p*-value	
G1	65 (28.5)	100 (41.3)	12 (16.4)	0.79	177
G2	93 (40.7)	73 (30.2)	34 (46.6)	0.84	200
G3	35 (15.4)	33 (13.6)	14 (19.2)	0.70	82
Unknown	35 (15.4)	36 (14.9)	13 (17.8)	—	84

**Table 3 cancers-17-02686-t003:** Reggio Emilia Cancer Registry. Distribution of endometrial cancer treatments across the three diagnosis periods.

	2017–2019	2020–2022	2023		Total
	*n* (%)	*n* (%)	*n* (%)	*p*-Value	*n*
Number of cases	228	242	73		543
Laparotomy	69 (30.3)	28 (11.6)	8 (11.0)	<0.01	105
Laparoscopy	137 (60.1)	189 (78.1)	59 (80.8)	<0.05	385
Vaginal hysterectomy	7 (3.1)	7 (2.9)	2 (2.7)	0.88	16
Laparotomy + laparoscopy	11 (4.8)	9 (3.7)	3 (4.1)	0.66	23
Unknown	4 (1.7)	9 (3.7)	1 (1.4)	0.70	14

**Table 4 cancers-17-02686-t004:** Reggio Emilia Cancer Registry. Distribution of time to surgery (TTS) for endometrial cancer across the three diagnosis periods, excluding patients whose date of diagnosis coincided with the date of treatment (131 cases, 6%).

	2017–2019	2020–2022	2023	
	Mean (SD)	Mean (SD)	Mean (SD)	
TTS, days	49 (50)	60 (51)	74 (98)	
TTS, *n*. cases	*n* (%)	*n* (%)	*n* (%)	*p*-value
1–30 days	23 (10.1)	9 (3.7)	1 (1.4)	<0.01
31–60 days	88 (38.6)	47 (19.4)	5 (6.9)	<0.01
61–90 days	52 (22.8)	72 (29.8)	27 (37.0)	<0.05
>90 days	17 (7.5)	56 (23.1)	15 (20.5)	<0.01
Total	228	242	73	

**Table 5 cancers-17-02686-t005:** Multiple regression analysis: estimated change in time to surgery (TTS, in days) by age, diagnosis period, and cancer stage.

	β	*p*-Value	CI 95%
Age	0.92	<0.01	0.38; 1.44
Diagnosis period	22.79	<0.01	14.67; 30.91
Stage	9.87	<0.05	3.55; 16.19
Age	0.89	<0.01	0.36; 1.42
2020–2022 (ref)	1		4.55; 10.12
2017–2019	−19.63	<0.01	−31.31; −7.95
2023	31.60	<0.01	13.68; 49.53
Stage I (ref)	1		
Stage II	5.24	0.74	−25.40; 35.87
Stage III	6.21	0.49	−11.29; 23.71
Stage IV	48.80	<0.01	23.15; 74.45

## Data Availability

The data presented in this study are available on request from the corresponding author. The data are not publicly available due to ethical and privacy issues; requests for data must be approved by the Ethics Committee after the presentation of a study protocol.

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
