# Peer review of "Impact of COVID-19 on the Stage and Treatment of Endometrial Cancer: A Cancer Registry Analysis from an Italian Comprehensive Cancer Center"

_cancers, 2025, doi:10.3390/cancers17162686_

Round 1

Reviewer 1 Report

Comments and Suggestions for Authors

The authors present a retrospective registry-based study evaluating the impact of the COVID-19 pandemic on the stage, treatment modalities, and surgical waiting times for patients with endometrial cancer at a single comprehensive cancer center in Reggio Emilia, Italy. They analyzed data across three time periods (pre-pandemic, pandemic, post-pandemic) focusing on changes in cancer stage distribution, surgical methods, and delays in surgery.

Although clearly presented, the study addresses a widely recognized and extensively documented phenomenon—that the COVID-19 pandemic resulted in delays to cancer treatments globally. The findings, such as increased delays in surgery and a slight shift towards higher-stage diagnoses, confirm outcomes that are already well-established in literature and common clinical understanding. As such, the manuscript does not significantly advance current scientific knowledge or propose new insights or solutions to mitigate these effects. The simple observational nature, confined to a single-center dataset, further limits the broader applicability and novelty of the findings. Therefore, this manuscript lacks the scientific impact and originality required for publication in the journal.

Author Response

Comment:s:

Although clearly presented, the study addresses a widely recognized and extensively documented phenomenon—that the COVID-19 pandemic resulted in delays to cancer treatments globally. The findings, such as increased delays in surgery and a slight shift towards higher-stage diagnoses, confirm outcomes that are already well-established in literature and common clinical understanding. As such, the manuscript does not significantly advance current scientific knowledge or propose new insights or solutions to mitigate these effects. The simple observational nature, confined to a single-center dataset, further limits the broader applicability and novelty of the findings. Therefore, this manuscript lacks the scientific impact and originality required for publication in the journal.

Response

Thanks to the reviewer for the comment. We agree with the reviewer that the article does not revolutionize the scientific world, but what we intend to do is observe, in a context of northern Italy that had a strong impact in terms of hospitalizations and deaths during the pandemic, what the impact has been on a pathology that has not been widely explored in the literature.

As another reviewer noted, perhaps the fact that gynecologists changed their approach during the pandemic (performing a greater number of laparoscopies compared to laparotomies), even in older women and with more advanced tumors, could justify publishing this work. Although it refers to a single center, the results are certainly generalizable, at least to northern Italy, characterized by a healthcare system with very similar diagnostic and therapeutic approaches. We have added a note in the conclusions.

Reviewer 2 Report

Comments and Suggestions for Authors

This manuscript presents a population-based study on the impact of the COVID-19 pandemic on the diagnosis and treatment of endometrial cancer in an Italian comprehensive cancer center. The study uses data from the Reggio Emilia Cancer Registry from 2017 to 2023, comparing pre-COVID, COVID, and post-COVID periods. The findings regarding increased advanced-stage cancers and prolonged time to surgery are important and timely. The manuscript is well-structured and the use of a population-based registry adds significant value.

Comments to Authors:

Abstract Section:
The abstract is largely suitable and well-structured, as it successfully conveys the study's purpose, methods, key findings, and conclusion. It is well-written and provides a concise overview of the paper's content.
1) Lines 46-47: All keywords should be verified according to MeSH (Medical Subject Headings). Additionally, I recommend adding ‘Italy’ as a keyword, as it is the location of your study.

Introduction Section:
2) The introduction could more explicitly define what "post-COVID" means in the context of this study. While the abstract and methods state it's 2023, the introduction could clarify that this period represents a time of recovery and normalization of healthcare services. This would help frame the later finding that delays persisted even after the pandemic's peak.
However, I recommend the current article for the introduction section:
DOI: 10.1155/sci5/6840605. 
- Lines 77-78: The aim of the study is clearly stated: to assess the pandemic's impact on the management of EC and determine if it compromised oncological outcomes.

Materials and Methods Section:
3) Lines 102-105: The current section mentions that TTS was calculated as the interval between diagnosis and surgery. However, it doesn't clearly define what event constituted "diagnosis" (e.g., initial symptom, first biopsy, pathology report confirmation). This is a critical detail, especially when discussing delays, as it could impact how the results are interpreted and compared to other studies.
- Lines 81-85: Data Source: The use of the Reggio Emilia Cancer Registry (RE-CR) is a major strength, as it's a high-quality, population-based registry with a high percentage of microscopic confirmation.
- Lines 100-101: Study Period: The division into three distinct periods (pre-COVID, COVID, and post-COVID) is appropriate for the study's objective.
- Lines 102-111: Data Analysis: The methods for analyzing data, including the use of multiple regression analysis to identify factors influencing time to surgery (TTS), are appropriate and well-described.

Results Section: 
4) Line 24 ,123 & 172: About Statistical significance of Grade 3 tumors, the authors mention that in the abstract and results section state that a "slight increase was noted in grade 3 tumors" , and the discussion says it was "not statistically significant". However, Table 2 shows the p-value for the grade distribution is 0.70. While not a gap, a more explicit sentence in the results section stating "The increase in grade 3 tumors was not statistically significant (p=0.70)" would prevent any ambiguity.
5) Lines 132-138: Clarity on exclusions: The methods section mentions excluding cases where the surgery date coincided with the diagnosis date but doesn't state how many were excluded. It's later noted in Table 4's caption that 131 cases were excluded, which could be stated upfront in the text for clarity.

Discussion Section:
6) The current text, the authors mention that links the delays to hospital reorganization and patient fear. To make this more robust, the authors could elaborate on how these factors specifically contributed to the TTS delays observed, especially the significant increase in the post-COVID period. 
I recommend the authors refer to the following article in this section for more details:  
Jafari et al. (2025) Unraveling the Intricacies of Gut Microbiome, Psychology, and Viral Pandemics: A Holistic Perspective. *Jordan J. Biol. Sci*, Vol. 18, Issue 1."

Conclusion Section:
7) Lines 248-252: The conclusion suggests the need for "control measures". However, the paper doesn't provide specific, actionable recommendations based on its findings. For instance, what kind of control measures could mitigate future delays for older patients or those with advanced-stage disease? Expanding on this point would increase the practical impact of the study.

Author Response

Comments 1: The abstract is largely suitable and well-structured, as it successfully conveys the study's purpose, methods, key findings, and conclusion. It is well-written and provides a concise overview of the paper's content. 1) Lines 46-47: All keywords should be verified according to MeSH (Medical Subject Headings). Additionally, I recommend adding ‘Italy’ as a keyword, as it is the location of your study

Response 1: Thanks to the reviewer. We have corrected the keywords and added the word "Italy."

Comments 2:  The introduction could more explicitly define what "post-COVID" means in the context of this study. While the abstract and methods state it's 2023, the introduction could clarify that this period represents a time of recovery and normalization of healthcare services. This would help frame the later finding that delays persisted even after the pandemic's peak.

Response 2:  Thanks for the suggestion. Indeed, the COVID years have seen significant variability in Italy as well, with some regions feeling the impact of the pandemic early and experiencing faster recovery times for cancer screening and diagnostic and therapeutic modalities. We have clarified this further in the text. The recommended reference has been added in the discussion.

Comments 3:  Materials and Methods Section: 3) Lines 102-105: The current section mentions that TTS was calculated as the interval between diagnosis and surgery. However, it doesn't clearly define what event constituted "diagnosis" (e.g., initial symptom, first biopsy, pathology report confirmation). This is a critical detail, especially when discussing delays, as it could impact how the results are interpreted and compared to other studies.

Response 3: We thank the reviewer for this clarification. Indeed, as members of a Cancer Registry, we assume the diagnosis date is understood, but we have specified it clearly in the text. International registry rules require the first definitive histological report to be used as the diagnosis date (incidence date); in its absence, the hospital admission date is used.

Comments 4:  Results Section: 4) Line 24 ,123 & 172: About Statistical significance of Grade 3 tumors, the authors mention that in the abstract and results section state that a "slight increase was noted in grade 3 tumors" , and the discussion says it was "not statistically significant". However, Table 2 shows the p-value for the grade distribution is 0.70. While not a gap, a more explicit sentence in the results section stating "The increase in grade 3 tumors was not statistically significant (p=0.70)" would prevent any ambiguity.

Response 4: Thanks to the reviewer for the clarification. We've modified the text as suggested, and it actually seems more understandable now.

Comments 5:  Results Section:  5) Lines 132-138: Clarity on exclusions: The methods section mentions excluding cases where the surgery date coincided with the diagnosis date but doesn't state how many were excluded. It's later noted in Table 4's caption that 131 cases were excluded, which could be stated upfront in the text for clarity.

Response 5:  Thank you for your clarification. Indeed, as reported above, the diagnosis date (date of incidence) in some cases, in the absence of other information, coincides with the surgery date. These cases are excluded from the TTS analysis (we have specifically stated this in the text).

Comments 6: Discussion Section: 6) The current text, the authors mention that links the delays to hospital reorganization and patient fear. To make this more robust, the authors could elaborate on how these factors specifically contributed to the TTS delays observed, especially the significant increase in the post-COVID period. 

Response 6: Thank you for your clarification. The literature has widely reported how COVID-19 has also impacted patients' attitudes. Fear of hospitalization has, in some cases, reduced the opportunity for early diagnosis. We have added some references for discussion:  

Ghapanvari F, Namdar P, Moradi M, Yekefallah L. Psychological Impact of the COVID-19 on Hospitalized Patients: A Qualitative Study. Iran J Nurs Midwifery Res. 2022 Mar 14;27(2):92-98. doi: 10.4103/ijnmr.ijnmr_382_20. PMID: 35419267; PMCID: PMC8997181.

Jafari et al. (2025) Unraveling the Intricacies of Gut Microbiome, Psychology, and Viral Pandemics: A Holistic Perspective. *Jordan J. Biol. Sci*, Vol. 18, Issue 1."

Jafari AS, Mozaffari Nejad AS, Faraji H, Abdel-Moneim AS, Asgari S, Karami H, Kamali A, Kheirkhah Vakilabad AA, Habibi A, Faramarzpour M. Diagnostic Challenges in Fungal Coinfections Associated With Global COVID-19. Scientifica (Cairo). 2025 May 7;2025:6840605. doi: 10.1155/sci5/6840605. PMID: 40370518; PMCID: PMC12077979.

Comments 7: Conclusion Section: 7) Lines 248-252: The conclusion suggests the need for "control measures". However, the paper doesn't provide specific, actionable recommendations based on its findings. For instance, what kind of control measures could mitigate future delays for older patients or those with advanced-stage disease? Expanding on this point would increase the practical impact of the study.

Response 7:  Thanks to the reviewer. Indeed, at the end of the study, we highlighted some critical issues that had emerged in our hospital, but also the possibility of changing our approach thanks to what emerged during the COVID emergency. The study results were discussed with surgical oncologists, who appreciated the potential of the emergency period to change some diagnostic and therapeutic approaches (e.g., greater use of laparoscopy).

Reviewer 3 Report

Comments and Suggestions for Authors
  1. Line 55, 67, 124, 170, 175, 176, 179, 182, 185, 206, 213,228 - To be statistical correct, whenever mentioned the word “significant” – please report the corresponding p-value to support such statement with the word of “significant”. If no p-value could be reported – suggest replacing with other words or phrases.

  1. Race-ethnicity, Charlson comorbidity index or Elixhauser comorbidity score and prior (like 6 months prior) health-care usages frequency (inpatients visits, emergency visits, outpatient/virtual-care visits) were all related to health outcome measures. These important covariates could all be easily generated using health administrative-level data. Authors could consider adding race-ethnicity, prior comorbidity index and prior health-care usages frequency into the models.  If author has difficulty to generate these important covariates and apply them into the models – then this could be listed as one of the study limitations.

  1. Line 246: “An important limitation is that this is the work of a single cancer registry; therefore, the results of the study cannot be generalized.” - If this is the case, what is the scientific value/importance of publishing this paper?

  1. Study limitations only had one sentence – too few.

Author Response

Comments 1: Line 55, 67, 124, 170, 175, 176, 179, 182, 185, 206, 213,228 - To be statistical correct, whenever mentioned the word “significant” – please report the corresponding p-value to support such statement with the word of “significant”. If no p-value could be reported – suggest replacing with other words or phrases.

Response 1: Thanks to the reviewer for the comment. We've reviewed all the sentences with the references, correctly reporting the statistically significant wording and the p-value.

Comments 2: Race-ethnicity, Charlson comorbidity index or Elixhauser comorbidity score and prior (like 6 months prior) health-care usages frequency (inpatients visits, emergency visits, outpatient/virtual-care visits) were all related to health outcome measures. These important covariates could all be easily generated using health administrative-level data. Authors could consider adding race-ethnicity, prior comorbidity index and prior health-care usages frequency into the models.  If author has difficulty to generate these important covariates and apply them into the models – then this could be listed as one of the study limitations.

Response 2: Thanks to the reviewer for their comment. Obviously, it would be important to have this information to better understand the study results. Unfortunately, the data of comorbidity is not accessible to the Cancer Registry, and using different databases would require a new study protocol with the approval of the Ethics Committee. We are adding this missing information within the study limits and hope to include this among the variables to be investigated in the near future. Information on rece-ethnicity is not available because in Italy these are small minorities that do not allow for appropriate studies.

Comments 3:  Line 246: “An important limitation is that this is the work of a single cancer registry; therefore, the results of the study cannot be generalized.” - If this is the case, what is the scientific value/importance of publishing this paper?

Response 3: The reviewer made an appropriate comment. In fact, the study refers to a single center in Italy. However, during the COVID emergency, the level of hospitalizations and deaths was high in northern Italy, and in general, all levels of care, participation in cancer screening, diagnostics, and treatment are very similar across northern Italy. The regions of northern Italy share the same incidence of cancer and the same survival rates. Therefore, the study results can easily be extended to all areas of northern Italy characterized by the same levels of care.

Comments 4:  Study limitations only had one sentence – too few.

Response 4:  We agree with the reviewer: we have added some limitations of the study, including the one mentioned above.

Reviewer 4 Report

Comments and Suggestions for Authors

This paper is an interesting addition to the gowing literature on COVID impact on cancer services. The analysis is focused on endometrial cancer, which is relevnat because it is not so commonly analysed.

I have a couple of questions to be addressed, or perhaps clarifications:

  • the data analysed is from the cancer registry and the clinical data comes from a Cancer Centre. The question is all cases in the Reggio Emilia were performing diagnosis and surgery in this hospital. Put in another way, are all cases diagnosed in the region included in the analysis? It is not clear to me as it is described now and it is relevant to clarify. A complememtary question is if there were no private practice patients receivig treatment in private facilities.
  • the periodification of  COVID epidemic is always controversial (particularlt, regarding the 2022). I think it would be helpful to suport the period chosen with background information about its rationale. A possible suggestion would be to modify the period and assess the impact regarding 2022.

An interesting additional data may be the endometrial incidence age-adjusted rate by year (not period) to see if the cases fluctuaded in a specific direction during the COVID years and then discuss the potential of underdiagnosis at the peak of the pandemia

It seems that the pandemia was like an accelerator of the change of practice (laparoscopy vs laparotomy). This is interesting and deserve future confirmation. The fact that apparently all data comes from a single cancer centre makes this change more intrigguing. 

Author Response

Comments 1: Comments and Suggestions for Authors. This paper is an interesting addition to the gowing literature on COVID impact on cancer services. The analysis is focused on endometrial cancer, which is relevnat because it is not so commonly analysed. I have a couple of questions to be addressed, or perhaps clarifications: the data analysed is from the cancer registry and the clinical data comes from a Cancer Centre. The question is all cases in the Reggio Emilia were performing diagnosis and surgery in this hospital. Put in another way, are all cases diagnosed in the region included in the analysis?

Response 1: Thanks to the reviewer for the comment. The study includes all endometrial infiltrating tumors registered at the Cancer Registry between 2017 and 2023. All cases were included in the analyses (there are cases with missing information), since it is a population-based study.

Comments 2:  It is not clear to me as it is described now and it is relevant to clarify. A complememtary question is if there were no private practice patients receivig treatment in private facilities. 

Response 2: Thank you for your request for clarification. In the Reggio Emilia Cancer Registry area (northern Italy), there are few private clinics, while the public system is very active, providing free care to all residents, including foreigners. The few private clinics present are, however, affiliated with the National Health System: therefore, the Cancer Registry receives the same information (morphology reports and discharge forms) as patients admitted to public facilities (which in our case account for approximately 99%).

Comments 3:  The periodification of  COVID epidemic is always controversial (particular, regarding the 2022). I think it would be helpful to support the period chosen with background information about its rationale. A possible suggestion would be to modify the period and assess the impact regarding 2022.

Response 3: Thanks to the reviewer for the clarification request. The period investigated (2017-2023) was divided into three periods: the first (2017-2019) corresponds to the pre-COVID period; the second (2020-2021) corresponds to the COVID period that in our province recorded the greatest impact in terms of missed diagnoses and deaths. In fact, in addition to the lockdown during the months of March-June 2020, in our province we had a second wave of COVID at the end of the year, with effects recorded in 2021. The period 2022-2023 corresponds to the almost complete resumption of activities, called post-COVID.

Comments 4:  An interesting additional data may be the endometrial incidence age-adjusted rate by year (not period) to see if the cases fluctuaded in a specific direction during the COVID years and then discuss the potential of underdiagnosis at the peak of the pandemia.

Response 4: Thanks for the clarification. We actually divided the period into three subperiods to emphasize the impact of COVID, but we also added, as suggested by the reviewer, a trend in the incidence of endometrial cancer by year in our province, to better understand the year-by-year effects.

Comments 5: It seems that the pandemia was like an accelerator of the change of practice (laparoscopy vs laparotomy). This is interesting and deserve future confirmation. The fact that apparently all data comes from a single cancer centre makes this change more intrigguing.

Response 5: Thanks to the reviewer for the suggestion. We've actually integrated the final section, adding this information.

Round 2

Reviewer 1 Report

Comments and Suggestions for Authors

The authors have addressed the significance of the study in updated manuscript, and stated their observation clearly.

Reviewer 3 Report

Comments and Suggestions for Authors

Accept author's modifications in the manuscript.